# Supportive Care Needs of Newly Diagnosed Cancer Patients in a Comprehensive Cancer Center: Identifying Care Profiles and Future Perspectives

**DOI:** 10.3390/cancers16051017

**Published:** 2024-02-29

**Authors:** Loula Papageorgiou, Jean-Bernard Le Provost, Mario Di Palma, Marc Langlois, Israa Salma, Manuella Lopes, Etienne Minvielle, Maya Abbas, Florian Scotté

**Affiliations:** 1Patient Pathway Division, Gustave Roussy, 94805 Villejuif, France; loula.papageorgiou@gustaveroussy.fr (L.P.); mario.dipalma@gustaveroussy.fr (M.D.P.); israa.salma@gustaveroussy.fr (I.S.); manuella.lopes@gustaveroussy.fr (M.L.); may.abbas@gustaveroussy.fr (M.A.); 2Psycho-Oncology Unit, Gustave Roussy, 94805 Villejuif, France; jeanbernard.le-provost@gustaveroussy.fr; 3Doo Conseil, 33000 Bordeaux, France; marc.langlois@doo-conseil.com; 4i3-CRG, École Polytechnique, CNRS, Institut Polytechnique de Paris, 91762 Palaiseau, France

**Keywords:** supportive care, patient needs, patient pathway, personalized care

## Abstract

**Simple Summary:**

A retrospective study conducted by collecting data from the case consultation and medical records of 6217 newly diagnosed patients of various cancers at a comprehensive cancer center in France led to a definition of patients’ supportive care needs. The top five interventions were dietary (for 60% of patients), physiotherapy (33%), psychology (29%), social care (28%), and pain management (16%), with specific additional needs per subgroup of cancer. The aforementioned data suggest that early, multidisciplinary supportive care interventions should be required.

**Abstract:**

The prompt introduction of supportive care for patients with cancer leads to a better quality of life, potential survival benefits, and improvements in treatment safety. Considering that patients’ needs vary, descriptive assessments could serve as a compass for an efficient and prompt healthcare response. The aim of this study was to identify supportive care needs in newly diagnosed patients according to cancer type. A retrospective study was conducted by collecting data from the case consultation and medical records of a comprehensive cancer center in France. Patients’ needs were divided into twelve domains: nutrition, psychological support, psychiatric support, social care, physiotherapy, addictology, pain management, palliative care, pharmacology, complementary and alternative practice (CAM), sexual health, and speech therapy. Out of 6217 newly diagnosed patients of various cancer types who sought medical care at Gustave Roussy in 2021, 2541 (41%) required supportive cancer care (SCC), and of them, 1331 patients (52%) required two or more different SCC specialist interventions. The top five interventions were dietary (for 60% of patients), physiotherapy (33%), psychology (29%), social care (28%), and pain management (16%). Subgroup analysis according to cancer department highlighted additional specific needs: CAM for breast cancer patients (11%), speech specialist (27%) and addictologist (22%) interventions for ENT patients, psychiatry consultations for neurological patients (16%), and palliative care for dermatology patients (23%). The aforementioned data suggest that an early, multidisciplinary supportive care intervention should be required. Assembling human resources at the time of diagnosis within a dedicated day unit would be the next appropriate step in developing personalized care pathways related to the highlighted needs.

## 1. Introduction

Supportive care in cancer, as defined by the MASCC (Multinational Association of Supportive Care in Cancer), is the prevention and management of the physical and mental complications of cancer and its treatments, throughout the course of the disease and during post-treatment care [1]. Their definition, as published in the circular of 22 February 2005, proposed by the French Association for Supportive Care in Cancer (AFSOS), is “all the care and support necessary for patients, throughout illness, along with specific treatments, when any exist” [2]. Supportive care consists of the coordination of a multidisciplinary approach, involving, in a non-exhaustive way, psycho-oncology, onco-psychiatry, addictology, pain management, nutrition, palliative care, social services, physiotherapy, and non-drug complementary therapies (mediation, sophrology, etc.).

Supportive care has shown its efficacy in increasing quality of life for advanced-stage patients with prognoses of survival of less than one year [3,4].

A few recent studies based on this approach have suggested a benefit in terms of quality of life during the first months of treatment, and have also demonstrated a reduction in hospitalizations and emergency care visits, improved chemotherapy tolerance, and longer survival during the first year after diagnosis [5,6]. Supportive care needs vary according to cancer type and further factors such as age, gender, marital status, etc. [7,8].

Early introduction of palliative care for patients with metastatic non-small cell lung cancer contributes to prolonged survival and clinically meaningful improvements in quality of life and mood [9]. The ENABLE II trial showed that patients with advanced-stage cancer receiving early telemedicine-assisted palliative care had improved one-year survival as opposed to patients who entered the program three months later [10]. Moreover, a recent observational study highlighted a survival benefit for patients on immune checkpoint inhibitors in whom related adverse effects were detected early [11].

Our institution places great importance on supportive care in cancer (SCC), which is provided by a dedicated division (Patient Pathway Division) and divided into specialist units. Particular focus is given to psycho-oncology, pain control, physiotherapy, palliative care, nutrition, addictology, management of toxicities, and integrative medicine throughout the program ‘’ Living better with cancer”. At present, SCC is introduced at various time points, after oncologist request, as part of the standard care plan.

Based on previously mentioned studies and everyday clinical experience, we hypothesized that early systematic symptom identification prior to initiating specific oncological treatments could contribute to maintaining patients’ quality of life. This detection should be based on a structured system which is easily available and can lead to individualized, prompt specialized guidance. In order to define further steps, it is always very useful to look back at previous hospital experience, which can serve as a compass for an efficient and prompt healthcare response.

The aim of the present study was to identify the most frequent supportive care needs of patients suffering from cancer in each oncology department, as these are reflected in specialist supportive care interventions. 

## 2. Materials and Methods

This is a single-center, descriptive retrospective study conducted at Gustave Roussy. The analysis involved supportive care specialist consultations for all new patients being cared for due to cancer between the 1 January 2021 and 31 December 2021.

The study protocol was approved by the local ethics committee of the participating center according to the Declaration of Helsinki, 1964. In alignment with the hospital’s policy for every new patient, all patients signed a non-opposition document stating that their hospital data could be used for research purposes.

### 2.1. Inclusion and Exclusion Criteria

The inclusion criteria were the following: (1) patients with newly diagnosed cancer, either primary or secondary; (2) age ≥ 18 years old; (3) provided written non-opposition for personal and medical data analysis at the time of medical file creation.

Exclusion criteria were: (1) the inability to understand medical information and (2) the absence of written non-opposition.

### 2.2. Hospital Oncology Departments

New patients received care at the center’s eleven oncology departments: (1) ear, nose, and throat, (2) dermatology, (3) gastroenterology, (4) gynecology, (5) hematology, (6) breast cancer, (7) pulmonology, (8) neuro-oncology, (9) sarcomas/complex tumors, (10) urology, and (11) the department of early drug development (Département de l’Innovation Thérapeutique et des Essais Précoces, DITEP). The care was focused on targeted therapies for all previously mentioned cancer types. Due to the variability of cancer types and treatments within the department, patient data from DITEP were not included in the present publication.

### 2.3. Supportive Care Needs

Patients’ needs in supportive care were divided into twelve domains: (1) nutrition, (2) psychological support, (3) psychiatric support, (4) social care, (5) physiotherapy, (6) addictology, (7) pain management, (8) palliative care, (9) pharmacology, (10) complementary and alternative practice (CAM), (11) sexual health, and (12) speech therapy.

### 2.4. Data Collection and Analysis

Data were collected from the digital patient files (Winsimbad^®^) and extracted to Microsoft Excel 2016^®^ for statistical analysis.

## 3. Results

### 3.1. Patient Characteristics

A total of 6217 newly diagnosed patients of various cancer types sought medical care at Gustave Roussy in 2021. The five most represented departments were breast cancer (26% of patients), ENT (14%), gastroenterology (9.6%), gynecology (9.4%), and urology (8.6%), followed by pulmonology (8.3%), hematology (7.6%), endocrinology (6.3%), dermatology (5.6%), and, lastly, neurology.

### 3.2. Supportive Care Needs

Among new patients, 2581 (female (n = 1401)/male (1137) ratio = 1.23) sought supportive care within 78 ± 81.32 days. Forty-two percent of these patients visited at least two different supportive care specialists. Among patients referred to supportive care, two-thirds visited a nutritionist expert and one-third met a physiotherapist. Around one-third of patients sought psychologist care and social care, and 16% of patients were referred for pain management consultations.

When looking into results of the main subgroups, there were specific SCC ‘’profiles’’ that emerged, reflecting the nature of the disease and the specificities of each population. As an example, patients—mostly women—with breast cancer, which often present fatigue, alterations in self-image, and stress, were in need of physiotherapy, psycho-oncology, social assistance, complementary care, and pain management. As for patients with ENT cancer, who often described dysphagia and anorexia as the main symptoms, nutritional consultation was the primary need.

Data on the top five supportive care interventions for all patients per department are shown in Table 1.

## 4. Discussion

Usually, cancer patients are referred for SCC with delays when presenting severe complications such as uncontrolled pain, depression, or cachexia, or during emergency hospitalizations. This oncology- and late-symptom-onset-led approach prevents them from benefiting from prompt and personalized SCC interventions. It is a proven fact that anticipating SCC can lead to minimized weight and muscle loss and changes in one’s psychological status and social life, as well in one’s financial burden. Overall, prompt assessment of and response to supportive care needs are positively related to patient autonomy and quality of life.

The French Cancer Institute (INCa) states, in its guidelines, that SCC should begin to be provided early and should continue throughout the course of the disease and after cancer treatment [12].

The present assessment is a retrospective study conducted on a substantial number of newly diagnosed patients of various cancer types. At least half of this large patient population was referred to supportive care specialists, with one-fourth of them needing two or more specific interventions within the first year of diagnosis. This finding is in alignment with other studies. A study on 173 women with newly diagnosed breast cancer showed that 44% of them presented with unmet supportive care needs at the time of diagnosis [13]. Among 136 pancreatic cancer patients, 96% of them reported at least one unmet supportive care need, mainly physical or psychological [14].

Subgroup analysis for each department further highlighted specific needs. Eleven per cent of breast cancer patients were referred to complementary practice teams due to a need for anti-stress and exercise methods. Around 27% of ENT patients, very often presenting articulation issues and tobacco and/or alcohol abuse problems, visited speech therapists, and 21% visited addictologists. Neurology patients experiencing rapid declines in autonomy and morale were in more frequent need of social assistance (64% of patients) and psychological or psychiatric intervention (35% and 16%, respectively) than others. Dermatology patients, mostly being treated for metastatic melanomas, were referred more frequently to palliative care (23% of patients) than other patients. Data are shown in Table 1.

The current study demonstrates that there was a slight predominance of women vs. men addressed in supportive care, on the order of 20%. Whether this is reflecting a general tendency or whether it correlates with specific interventions is to be investigated in further studies.

Other investigators agree with the preventive approach with regard to supportive care. Timely integration of nutrition therapy and physical exercise emerges as a prerequisite for preventing side effects of therapy and reducing early-, mid-, and long-term toxicities [15]. Studies have proven that fatigue and muscle loss in breast cancer can be prevented by the early introduction of a physical exercise program for patients with breast and colon cancer [16,17].

The present study provides comprehensive evidence, in a large number of patients with various cancer types, that the majority of newly diagnosed cancer patients, irrespective of their sociodemographic profile, age, or cancer type, are in need of supportive care as early as throughout the first year after diagnosis.

A major limitation is that patient pathways were mainly traced at various time points by oncology practitioners who identified patient needs during consultation or hospitalizations and addressed patients accordingly. Therefore, there was not a common assessment tool applied for evaluating SCC needs.

A second limitation is that the identification of supportive care needs was based on the intervention of the supportive care teams, and, therefore, more on the identification of symptoms or clinical needs than on the identification of the patients’ day-to-day requirements. 

There is vast agreement that improving SCC delivery goals can only be achieved by adopting a stepping-stone approach through auto- and hetero-evaluation initiated at the time of new cancer diagnosis. Early and prompt auto-evaluation is an important contributor to precise assessment, as doctors often tend to downgrade symptoms [18,19]. Ideally, SCC should be delivered in dedicated multidisciplinary units.

In order to address these issues, and based on the institution’s experience, a program for early needs detection and introduction of supportive care was developed in our institute. An auto-evaluation digital multiple-choice questionnaire was developed by the multidisciplinary SCC department, enabling the detection of patient needs in all basic domains of supportive care.

To design this questionnaire, nine areas were identified by the supportive care teams of the patient pathways division, in conjunction with the hospital’s patient committee, in order to obtain an overall view of patients’ experiences and needs. Three validated questionnaires were included, covering the risks of poor adherence to treatment [20], anxiety/depressive symptoms (PHQ2) [21], and general symptoms (MD Anderson Symptom Inventory) [22]. According to the level of vulnerability, patients were given educational material about a healthy lifestyle or were offered appointments with an SCC specialist corresponding to their specific need. Some patients were also given appointments at the Evaluation Day Unit, where they met with a multidisciplinary team. This Evaluation Day Unit closed with a tailored-made care plan co-decided by the doctor and the patient. The purpose of this program is to identify patients with multiple needs, which can define complex situations. The self-assessment can be considered as a limit in several cancer diseases, like neurological tumors.

Further studies are to be conducted including patient auto-evaluation tools, which would cover even more aspects of everyday life and patient preferences. Supportive care is usually focused on physical or emotional needs. Concerns regarding spirituality and practical and social aspects are to be further assessed by using dedicated tools such as the NCCN distress thermometer [23]; and unmet needs using the CaSUN questionary [24]. Sexual health needs should also be evaluated [25,26]. Specific attention should be paid to evaluating caregivers’ needs [27,28,29]. Moreover, in the era of artificial intelligence, developing sophisticated digital tools for evaluation and follow-up based on large-scale real-life data [30] will enable a quicker, more personalized, and more prompt intervention of supportive care providers and ultimately contribute to improving the patients’ quality of life.

## 5. Conclusions

The aforementioned data suggest that early, multidisciplinary supportive care interventions should be required. Before any requirement, evaluation of patients’ needs is mandatory in order to correctly coordinate the necessary teams. Combining self-assessment might be the best way, with complementary assessment by the treating clinical team at the time of cancer diagnosis in order to trace the patient pathway. Assembling human resources early in the pathway, within a dedicated one-stop unit, would be the next appropriate step towards developing personalized care pathways related to the highlighted needs.

## Figures and Tables

**Table 1 cancers-16-01017-t001:** The top five supportive care interventions according to cancer type: breast cancer, ear nose throat (ENT), gastroenterology, gynecology, urology, pulmonology, hematology, endocrinology, dermatology, sarcoma and rare tumors, and neurology, including profiles for all new patients in our center for the year 2021.

Cancer Department	New Patients (n)	Patients Requiring Supportive Care Interventions (%)	1st Intervention	2nd Intervention	3rd Intervention	4th Intervention	5th Intervention
ENT	847	607 (72%)	Nutritionist (38%)	Physiotherapist (12%)	Social assistant (12%)	Speech therapist (11%)	Addictologist (9%)
Pulmonology	501	128 (29.5%)	Nutritionist (53%)	Social assistant (36%)	Psychologist (35%)	Pain specialist (24%)	Physiotherapist (24%)
Gastroenterology	580	322 (55.5%)	Nutritionist (40%)	Social assistant (13%)	Pain specialist (6%)	Physiotherapist (16%)	Psychologist (13%)
Dermatology	336	57 (17%)	Psychologist (20%)	Nutritionist (20%)	Social assistant (17%)	Palliative care (14%)	Pain specialist (11%)
Neurology	165	69 (42%)	Social assistant (37%)	Psychologist (20%)	Physiotherapist (13%)	Nutritionist (12%)	Psychiatrist (9%)
Gynecology	565	178 (31.5%)	Nutritionist (30%)	Psychologist (20%)	Physiotherapist (15%)	Social assistant (13%)	Pain specialist (7%)
Hematology	455	247 (54%)	Nutritionist (36%)	Psychologist (16%)	Social assistant (14%)	Pain specialist (14%)	Physiotherapist (6%)
Sarcoma	288	126 (44%)	Nutritionist (33%)	Physiotherapist (22%)	Social assistant (14%)	Psychology (11%)	Pain specialist (9%)
Breast cancer	1569	568 (36.2%)	Psychologist (22%)	Physiotherapist (36%)	Social assistant (12%)	CAM (7%)	Pain specialist (7%)
Urology	518	133 (25.6%)	Nutritionist (25%)	Psychologist (17%)	Social assistant (15%)	Pain specialist (15%)	Physiotherapist (11%)
Early drug development department (All cancer types)	131	61 (30%)	Nutritionist (64%)	Physiotherapist (16%)	Pain specialist (31%)	Social assistant (16%)	Psychologist (11%)
All patients	5955	2581 (43%) of which 1348 (52%) requiring 2 or more interventions	Nutritionist 1513 (59%)	Physiotherapist (33%)	Psychologist (14%)	Social assistant (14%)	Pain specialist (16%)

## Data Availability

Data is retained within this article.

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
