# Peer review of "Supportive Care Needs of Newly Diagnosed Cancer Patients in a Comprehensive Cancer Center: Identifying Care Profiles and Future Perspectives"

_cancers, 2024, doi:10.3390/cancers16051017_

Round 1

Reviewer 1 Report

Comments and Suggestions for Authors

The big picture idea of what the project can evolve into is novel, however the current data is simply an observation of "referrals" made to supportive care specialists. This is a simple metrics of proportion but does not help support the bold claims made in the conclusions such as "The aforementioned data suggest that an early, multidisciplinary supportive care  should be required"

Without a comparator arm, this conclusion is premature. 

Comments on the Quality of English Language

needs revision

Author Response

We thank the reviewer for the time spent to read and review our manuscript. The study presented in the manuscript is actually a "picture" of the referrals to supportive care specialists and gave metrics in order to give us a background on how to develop our program of early needs assessment. The different pathways built with SCC actors was impossible without the evaluation of contacts during the cancer course of each type of tumor.

We thought interesting to present our methodology to develop our program and to highlight the needs of patients in a one year cohort and showing the discrepancies between cancers. 

A comparator arm was impossible following that method. 

Reviewer 2 Report

Comments and Suggestions for Authors

This is well conducted and an important review that is a useful springboard for further areas of research and developments of supportive oncology care integrated into clinical practice.

One suggestion - on page 5, in the table, I would write 'breast cancer' and not Senology. Senology is not commonly used in English.

Comments on the Quality of English Language

Generally fine I have one suggestion - on page 5, in the table, I would write 'breast cancer' and not Senology. Senology is not commonly used in English.

Author Response

We thank the reviewer for his comment. We have changed sexology into breast cancer in the table and in the text of the manuscript.

Reviewer 3 Report

Comments and Suggestions for Authors

Thank you for the opportunity to review this interesting paper.

In general, the paper covers an important topic, but it is a pity that the applied supportive care offers were not analyzed more in-depth. Instead, the discussion is very long and exceeding the information gained in the specific study by far.

Here are some specific recommendations:

Throughout text: Consistently add a space before the % or not, check for multiple and unnecessary spaces (before commas, quotation marks and reference indices), attach headings to the following section.

Abstract:

l. 32: Supportive cancer care (SCC)?

l: 37: Introduction of ENT abbreviation missing

Introduction:

Content-wise the introduction is short but clear and leads to the point but, if available, a reference might also be given for the assumption that supportive care needs differ by cancer site and further factors, as this comparison is done in the study.

l. 47 ff: The verb in the sentence is missing (…”is the prevention”?).

l. 67: Shouldn’t it be later (for the control group) instead of earlier?

l. 69: “in whom” instead of “to whom”?

Methods:

l.101: Although the abbreviation ENT might be common, I would prefer to explain it when it is first mentioned

l. 102: gastroenterology (o missing)

l. 106: “To” before “Due to” is too much

l. 114: Microsoft Excel

Results:

l. 135: main symptoms?

Table 1: I wonder whether comparison would be facilitated by using a bar graph instead of a table, using different colors for the different supportive care interventions. But I would leave it up to the authors what they prefer.

l. 144 ff: The heading “correlates of supportive care needs” suggests an analysis which offers were often applied together, but the text is a repetition and interpretation of table 1, this part belongs to the discussion in my understanding. To show intercorrelations between the offers in a cross table would be very interesting (e.g. of those having physiotherapy, how many % also had pain management).

It would also be very interesting to do further analyses based on patients’ characteristics (age, sex, tumor type (on a more detailed level), stage, marriage status) if this data is available. This would give more insight into the reasons for the different needs of the groups and would give clinicians more of an idea of what a specific patient might need.

Discussion:

l. 183: Unfortunately, it was not analyzed whether the needs were irrespective of these factors.

l. 206 ff: The SCNS might be considered https://onlinelibrary.wiley.com/doi/10.1002/pon.1740, as well as the CASUN https://pubmed.ncbi.nlm.nih.gov/17177268/, although these instruments might not exactly cover what is needed but it seems that other people have already spent thoughts on the topic.

The descriptions of your own developments between l. 199 and 242 can be radically shortened, this is not the discussion of the results. I would be happy to review a paper that is specifically dedicated to the newly developed instrument and improvement of the pathway to supportive care in your clinic. This is a very important topic but in a scientific paper the discussion should interpret the specific results shown before and give some future outlook, not more.

Conclusion:

In l. 255 you write that self-assessment is best but this encompasses the problems mentioned in l. 226 ff. So in general, self-assessment might be the best way, with complementary assessment by the treating clinical team where needed.

Comments on the Quality of English Language

Overall, the English is fine but some wordings are a bit rare and I recommend to have the manuscript reviewed by a native English speaker.

Author Response

Authors would like to thank the reviewer for the time and effort spent and for the useful comments.

Here are some specific recommendations:

Throughout text: Consistently add a space before the % or not, check for multiple and unnecessary spaces (before commas, quotation marks and reference indices), attach headings to the following section

Answer : editing was checked as indicated

Abstract:

32: Supportive cancer care (SCC)?

Answer: ‘’supportive care’’ replaced by ‘’Supportive cancer care (SCC)  “

37: Introduction of ENT abbreviation missing: ‘’ interventions for Ear Nose and Throat (ENT) patients’’

Introduction:

Content-wise the introduction is short but clear and leads to the point but, if available, a reference might also be given for the assumption that supportive care needs differ by cancer site and further factors, as this comparison is done in the study:

Two references were added:

[7] Reilly CM, Bruner DW, Mitchell SA, et al. A literature synthesis of symptom prevalence and severity in persons receiving active cancer treatment. Support Care Cancer. 2013;21(6):1525-1550. doi:10.1007/s00520-012-1688-0

[8] Cuthbert CA, Boyne DJ, Yuan X, Hemmelgarn BR, Cheung WY. Patient-reported symptom burden and supportive care needs at cancer diagnosis: a retrospective cohort study. Support Care Cancer. 2020;28(12):5889-5899. doi:10.1007/s00520-020-05415-y

47 ff: The verb in the sentence is missing (…”is the prevention”?).

47; « Is » has been added: "Supportive care in cancer, as defined by MASCC (Multinational Association of Supportive Care in Cancer) is the prevention and management of physical and mental complications of cancer and its treatments, throughout disease and during post-treatment care »

67: Shouldn’t it be later (for the control group) instead of earlier?

« earlier » has been changed in « later » 

"The ENABLE II trial showed that patients with advanced stage cancer receiving early telemedicine-assisted palliative care, had an improved one-year survival as opposed to patients who entered the program three months later. »

69: “in whom” instead of “to whom”?

« to whom » has been changed in « In whom » ; Moreover, a recent observational study highlighted a survival benefit for patients on immune checkpoint inhibitors in whom related adverse effects were detected early.

Methods:

101: Although the abbreviation ENT might be common, I

would prefer to explain it when it is first mentioned :

Answer : replaced by « Ear Nose and Throat »

102: gastroenterology (o missing) :

 Answer : corrected  to «gastroenterology»

106: “To” before “Due to” is too much

Answer : Corrected : « Due to… »

114: Microsoft Excel

Answer : « Microsoft » added

Results :

135: main symptoms ?

Answer : currently on line 137 , corrected to « Main symptoms »

Table 1: I wonder whether comparison would be facilitated by using a bar graph instead of a table, using different colors for the different supportive care interventions. But I would leave it up to the authors what they prefer.

Answer : Thank you for this usefull suggestion. We would like to maintain the table, transformed into an editable form as suggested by the editor.

144 ff: The heading “correlates of supportive care needs” suggests an analysis which offers were often applied together, but the text is a repetition and interpretation of table 1, this part belongs to the discussion in my understanding.

Answer : Thank you for this usefull comment. The paragraph is introduced to the discussion, l166 -174.

To show intercorrelations between theoffers in a cross table would be very interesting (e.g. of those having physiotherapy, how many % also had painmanagement).It would also be very interesting to do further analyses based on patients’ characteristics (age, sex, tumor type (on a more detailed level), stage, marriage status) if this data is available. This would give more insight into the reasons for the different needs of the groups and would give clinicians more of an idea of what a specific patient might need.

Answer : Thank you for these accurate and usefull suggestions. We aim to  perform this analysis on  data that emerge from the newly develloped tool in our clinic.

Discussion:

183: Unfortunately, it was not analyzed whether the needs were irrespective of these factors.

206 ff: The SCNS might be considered https://onlinelibrary.wiley.com/doi/10.1002/pon.1740, as well as the CASUN

https://pubmed.ncbi.nlm.nih.gov/17177268/, although these instruments might not exactly cover what is needed but it seems that other people have already spent thoughts on the topic.

The descriptions of your own developments between l. 199 and 242 can be radically shortened, this is not the discussion of the results. I would be happy to review a paper that is specifically dedicated to the newly developed instrument and improvement of the pathway to supportive care in your clinic. This is a very important topic but in a scientific paper the discussion should interpret the specific results shown before and give some future outlook, not more.

Answer : Thank you , we limited this part in lines 204 to 215

Conclusion:

In l. 255 you write that self-assessment is best but this encompasses the problems mentioned in l. 226 ff. So in general, self-assessment might be the best way, with complementary assessment by the treating clinical team where needed.

Answer : Combining self-assessment might be the best way, with complementary assessment by the treating clinical team, at the time of cancer diagnosis in order to trace the patient pathway

Comments on the Quality of English Language Overall, the English is fine but some wordings are a bit rare and I recommend to have the manuscript reviewed by a native English speaker.

Answer : A native english speaker checked the manuscript.

Round 2

Reviewer 1 Report

Comments and Suggestions for Authors

Authors have made improvements to the manuscript. As such, the flow and clarity of manuscript is now good. 

Comments on the Quality of English Language

Should be edited for sentence structure by experts

Reviewer 3 Report

Comments and Suggestions for Authors

The authors have addressed my questions and comments. Table 1 looks like not fitting on the page now but I think this will be solved during the final layout and editing process.